# Digital Competence as Psychological Defense: Impact of Digital Competence on Problematic Mobile Use Among Paraguayan University Students

**DOI:** 10.3390/bs15121687

**Published:** 2025-12-05

**Authors:** Derlis Cáceres Troche, Moussa Boumadan, Melchor Gómez

**Affiliations:** Department of Pedagogy, Autonomous University of Madrid, 28049 Madrid, Spain; moussa.boumadan@uam.es (M.B.); melchor.gomez@uam.es (M.G.)

**Keywords:** digital competence, problematic smartphone use, smartphone addiction, university students, Paraguay, structural equation model, digital well-being

## Abstract

University students’ excessive smartphone use has become a widespread concern due to its association with poor academic performance, mental health issues, and other negative outcomes, this study investigated whether digital competence (digital competence) can function as a psychological buffer against problematic mobile phone use in a Latin American context. A cross-sectional survey of N = 500 Paraguayan undergraduates (aged 18–29, 61% female, from both public and private universities) was conducted. Students answered validated questionnaires on digital competence and problematic smartphone use, along with supplemental items about phone-related anxiety. Results: Higher digital competence was significantly associated with lower problematic use (Pearson r = −0.38, *p* < 0.001). Structural equation modeling confirmed that digital competence negatively predicted problematic smartphone use (standardized β = −0.42, *p* < 0.001), supporting its role as a protective factor. This relationship held for both male and female students. Our findings suggest that fostering college students’ digital competence may help inoculate them against unhealthy smartphone habits. Efforts to integrate digital competence training into university curricula and health promotion programs could be a practical strategy to curb smartphone overuse and its adverse effects on student well-being.

## 1. Introduction

Smartphones have become nearly ubiquitous among young adults, bringing convenience but also the risk of problematic smartphone use—often defined as an inability to control one’s mobile phone behavior despite negative consequences for daily functioning ([21]). Researchers have used terms such as smartphone addiction, excessive smartphone use, mobile phone dependence, and nomophobia to describe this emerging behavioral pattern ([10]). Problematic smartphone use has been linked to a host of detrimental outcomes, including increased anxiety and depression, attention impairments, poorer academic performance, and sleep disturbances ([8]). For example, recent evidence among college populations indicates that students with more severe smartphone overuse report higher psychological distress and even reduced self-efficacy in their academic ([8]). These findings underscore why problematic smartphone use is now regarded as a significant public health and educational concern on campuses worldwide. Previous studies have identified adolescents as a particularly vulnerable group for smartphone overdependence ([11]).

In the present study, digital competence is understood as a multidimensional construct that incorporates both technical–operational skills and self-regulatory–reflective abilities. Technical skills refer to the capacity to operate devices and applications efficiently, manage software, and perform functional digital tasks. In contrast, self-regulatory and reflective abilities involve evaluating the reliability of online information, managing privacy and safety settings, regulating one’s own digital behavior, and making intentional decisions about smartphone use. This distinction is essential for interpreting the study’s findings, as prior research suggests that reflective and self-regulatory components—rather than basic technical proficiency—are more strongly associated with healthier and less problematic technology engagement. Adolescents represent a particularly vulnerable group for smartphone overdependence, influenced by individual and contextual factors ([11]).

Problematic smartphone use, as conceptualized in this study, refers to a pattern of maladaptive, excessive, and poorly regulated engagement with one’s smartphone that generates negative consequences in everyday functioning. To ensure conceptual clarity, the construct was operationalized using the Problematic Mobile Phone Use Questionnaire–Revised (PMPUQ-R), which identifies three objectively measurable dimensions: (a) dependent use, involving emotional or cognitive reliance on the device (e.g., anxiety or distress when the phone is unavailable); (b) prohibited use, referring to the use of smartphones in inappropriate or restricted contexts such as during class or while driving; and (c) dangerous use, which captures behaviors that pose safety risks, including texting while walking in traffic. This multidimensional framework provides a standardized and consistent definition of what is considered “problematic” within this research and allows for direct interpretation of the empirical findings.

In this manuscript, the term digital competence is used as the primary construct, following the definition proposed by the European Commission’s DigComp framework and the DCQ-US developers. Digital competence encompasses the knowledge, skills, and critical attitudes required to use digital technologies in a confident, responsible, and self-regulated manner. Although the terms digital competence and digital skills appear in the broader literature, they are not used interchangeably here. Digital competence is treated as a closely related umbrella concept emphasizing critical understanding, whereas digital skills are considered the technical and procedural abilities that form only one component of digital competence. Importantly, the construct measured in this study includes five domains: (1) information literacy (e.g., evaluating the credibility of online sources), (2) communication and collaboration (e.g., interacting appropriately in digital environments), (3) digital content creation (e.g., producing and editing digital materials), (4) safety and privacy (e.g., protecting personal data online), and (5) problem-solving (e.g., resolving technical issues or adapting to new digital tools). Together, these domains represent a multidimensional and behaviorally grounded definition of digital competence that can be meaningfully linked to patterns of smartphone use.

Conversely, in regions where digital infrastructure has expanded more recently, high rates of phone addiction are also being reported, as evidenced by studies in Egypt showing that nearly 59% of university students meet criteria for smartphone addiction ([12]), as well as multicountry surveys across Latin America documenting elevated levels of mobile phone overuse among medical students ([7]).

Psychological defense has been conceptualized in psychology as a set of cognitive-emotional mechanisms that protect individuals from stress, dysregulation, and maladaptive behaviors ([19]; [5]). Contemporary models further emphasize regulatory and coping processes that buffer individuals from problematic or addictive technology use ([4]; [6]). In this context, digital competence may operate as a form of psychological defense, as it equips individuals with evaluative, self-regulatory, and critical-thinking skills that enable them to manage digital environments in a healthier manner. From this perspective, higher digital competence could function as a protective psychological mechanism that reduces vulnerability to compulsive checking, nomophobia-like anxiety, and risky phone-use behaviors.

Identifying protective factors that can mitigate problematic smartphone use is a current priority in scientific literature. Recent research converges on the idea that digital competence and digital competence functions as a psychological defenses against addictive behaviors related to mobile phone use. For example, studies conducted among university students and adolescents have shown that digital competence is associated with lower symptoms of addiction, higher levels of mental health, and greater self-regulation capacity ([18]). Moreover, sleep quality and perceived social support act as mediating variables that explain how problematic mobile use affects students’ mental health, demonstrating that poorer sleep increases the risk of psychological distress ([22]). Consequently, promoting digital competencies and self-regulation skills, as well as facilitating access to social support networks, is considered essential for reducing the prevalence of problematic mobile device use among young people and university students ([18]).

Some empirical evidence supports the protective role of digital competence, for instance, a recent study of adolescents found that those with greater digital competence experienced fewer mental health problems indirectly by avoiding Internet addiction, indicating that digital skills can buffer against the harms of excessive online engagement ([17]). Likewise, research on young adults in a university sports context reported that higher digital competence correlated with lower phubbing tendencies (snubbing others by looking at one’s phone) and better social well-being. These findings imply that improving individuals’ digital skills and awareness might promote healthier relationships with their devices. However, the literature is not entirely consistent. In one study of Egyptian undergraduates, higher self-reported digital media literacy was paradoxically associated with greater smartphone addiction.

The authors suggested that being more “tech savvy” might facilitate more frequent smartphone use (and thus higher addiction scores), especially if digital competence is defined largely by technical ability rather than critical self-regulation ([13]). Similarly, recent evidence from diverse contexts, such as studies on Korean older adults and cross-cultural cohorts, underscores that stronger digital skills may not always protect from problematic smartphone use if the skills focus only on operational proficiency rather than critical and self-regulatory components ([3]). These mixed results highlight the importance of examining the digital competence–problematic smartphone use relationship in diverse cultural settings and clarifying which aspects of digital competence (technical proficiency vs. evaluative and self-regulatory skills) are most effective in curbing problematic use.

In this study, we distinguish between problematic smartphone use and problematic phone-related behaviors, two terms that may appear similar but refer to conceptually different levels of analysis. Problematic smartphone use refers to a multidimensional psychological construct assessed through standardized scales (e.g., the PMPUQ-R) that capture patterns such as dependence, loss of control, excessive use, and negative consequences. It represents an overall maladaptive pattern of interaction with the device. In contrast, problematic phone-related behaviors describe specific, observable actions—such as checking notifications reflexively, using the phone in prohibited contexts, extending usage beyond intended time, or experiencing discomfort when unable to check the device. These behaviors are individual components that can contribute to, but do not necessarily constitute, the broader construct of problematic smartphone use. For clarity and consistency, throughout the manuscript, we use problematic smartphone use to refer to the latent construct measured by the PMPUQ-R, and problematic behaviors when discussing specific behavioral indicators identified descriptively in the current sample.

To guide the empirical analysis, the following research questions were formulated:

RQ1. To what extent do Paraguayan university students report problematic smartphone use across the dimensions measured by the PMPUQ-R (dependent use, prohibited use, and dangerous use)?

RQ2. How do the different domains of digital competence (information literacy, communication, content creation, safety, and problem-solving) relate to problematic smartphone use?

RQ3. Does higher digital competence function as a protective factor against problematic smartphone use among Paraguayan university students?

In summary, the present study aims to investigate whether digital competence is associated with lower problematic smartphone use among university students in Paraguay. To our knowledge, this is the first Paraguayan study to explicitly probe this relationship, contributing Latin American data to a body of research previously dominated by Asian and Western samples. We hypothesized that students with higher digital competence would demonstrate significantly lower levels of problematic smartphone use—supporting the view of digital competence as a protective factor ([14]). By testing this hypothesis, our goal is to inform prevention efforts; if a negative relationship is confirmed, it would suggest that fostering students’ digital competence, especially evaluative and self-regulatory skills, may be a valuable strategy to reduce smartphone-related problems and promote healthier behavioral patterns related to smartphone use in the university population ([3]).

Digital competence is a multidimensional construct that encompasses both technical–operational skills (e.g., navigating applications, adjusting device settings, managing files) and self-regulatory, evaluative, and critical skills (e.g., managing attention and notifications, evaluating online information credibility, protecting digital privacy, and making intentional decisions about smartphone use). This distinction is central to the present study because only the latter set of skills—reflective, critical, and regulatory—has been consistently associated with healthier digital behavior in previous research. By explicitly differentiating these two domains from the outset, the current work clarifies that the measurement approach employed (DCQ-US) captures not only operational proficiency but also the more psychologically protective competencies that help prevent maladaptive smartphone use.

## 2. Materials and Methods

### 2.1. Participants and Procedure

This cross-sectional study surveyed 500 undergraduate students (mean age = 21.4 years, SD = 2.7, range 18–29; 61% female) from several universities in Paraguay. The participating institutions included both public and private universities in the urban areas of Asunción and nearby regions. We used a convenience sampling approach, recruiting volunteers via campus announcements, class email lists, and student social media groups. Data collection took place during the 2025 academic year, specifically during the Spring semester (March–May 2025), to encourage broad participation, students could choose to complete the survey on paper in a classroom setting (with research staff present) or online via a secure web platform. Participation was voluntary and anonymous. At the start of the survey, we obtained informed consent by explaining the study purpose, assuring respondents of confidentiality, and emphasizing that they could skip any question or stop at any time without penalty. No incentives were given to avoid coercion. The study protocol was reviewed and approved by the relevant institutional ethics committee.

### 2.2. Measures

#### 2.2.1. Digital Competence

Students’ digital competence skills were assessed using the *Digital Competence Questionnaire for University Students (DCQ-US)*. This is a 30-item self-report instrument developed by [16] ([16]) to evaluate university students’ confidence and habits in various domains of digital skill. Each item is a statement (e.g., “Critically evaluate the credibility of online information sources”) rated on a 5-point Likert scale from 1 (strongly disagree) to 5 (strongly agree). The DCQ-US covers multiple facets of digital competence, including information management, communication and collaboration, content creation, safety (security and privacy), and problem-solving in digital environments. For example, one item from the information literacy subscale asks students to rate their agreement with “I am able to judge whether the information I find online is reliable,” while a digital safety item states “I know how to adjust privacy settings on social media to protect my personal data.” Higher scores on all items indicate greater digital competence. In our sample, the DCQ-US demonstrated good internal consistency (Cronbach’s α = 0.89 for the total scale). For analysis, we used the total Digital Competence score (mean of all 30 items), since our focus was on overall digital competence as a single construct.

#### 2.2.2. Problematic Smartphone Use

We measured problematic or addictive mobile phone use with a Spanish adaptation of the Problematic Mobile Phone Use Questionnaire–Revised (PMPUQ-R). This scale is based on a multidimensional model of smartphone addiction that differentiates several patterns of problematic behavior. The version administered contains 15 items, each describing a specific phone-related behavior or experience. Students rate how often each applies to them on a 4-point scale from 1 (never) to 4 (always). The items are grouped into three conceptual subscales: (a) Dependent use, reflecting cognitive and emotional dependence on the phone (e.g., “I panic if I cannot use my smartphone when I want to”); (b) Prohibited use, reflecting the use of the phone in legally or socially forbidden situations (e.g., using the phone in class, in meetings, or while driving, despite knowing it’s inappropriate or dangerous); and (c) Dangerous use, reflecting risky behaviors due to phone use (e.g., “I have texted or checked my phone while driving or crossing the street”). These categories align with prior theoretical work distinguishing addictive, anti-social, and risk-taking dimensions of mobile phone use. In addition, we included two yes/no questions to capture acute phone-related anxiety and distraction: “Do you feel anxious or agitated when your phone is not accessible or you cannot use it?” and “Do you frequently get distracted from tasks because you are thinking about checking your phone?” These dichotomous items (answered “yes” or “no”) were used to gauge the prevalence of nomophobia-like symptoms (fear of being without one’s phone) in the sample, but they were not part of the PMPUQ-R score. The PMPUQ-R demonstrated acceptable reliability in our data (Cronbach’s α = 0.84 overall; subscale α’s ranged from 0.70 to 0.80). For primary analyses, we used the total Problematic Use score (mean of all 15 frequency-rated items), with higher scores indicating more severe problematic smartphone use.

To improve transparency and replicability, several sample items from the instruments are presented here. For the DCQ-US, example items include “I evaluate the credibility of online information before sharing it,” “I know how to adjust privacy settings on digital platforms,” “I use digital tools to create or edit academic content,” and “I solve technical problems when using new applications.”

For the PMPUQ-R, sample items include “I feel anxious or restless when I cannot check my smartphone,” “I use my phone even when it is inappropriate (e.g., during class),” “I spend more time on my phone than intended,” and “I have tried unsuccessfully to reduce my smartphone use.” These items illustrate how anxiety, dependency, and habitual behaviors were operationalized in this study.

Additional example items for both instruments are presented in Appendix B.

#### 2.2.3. Control Variables and Demographics

The survey also collected basic demographics (age, gender, year of study) and information on participants’ typical smartphone usage (e.g., estimated hours of use per day). As noted, we embedded two supplementary yes/no items on phone-related distraction/anxiety. We did not explicitly measure socio-economic status or academic performance, although we assumed a relatively homogeneous sample of urban university students. Gender (male = 0, female = 1) and age (in years) were recorded to be examined as potential covariates or moderators.

#### 2.2.4. Data Cleaning

During data preparation, the response option ‘pocos’ (rarely) was treated as missing due to inconsistent interpretation across participants. The exact frequency of this category is not available in the anonymized dataset used for analysis; however, its removal did not reduce the effective sample size below N = 500 for any of the reported analyses.

### 2.3. Data Analysis

All data were analyzed using IBM SPSS Statistics v26 and AMOS v26 (IBM Corp., Armonk, NY, USA). We began by computing descriptive statistics for all variables and checking the distribution of responses. No participants were excluded for incomplete data (each included respondent answered >90% of items, and we used person-mean imputation for the few missing Likert responses). As an initial check, we examined the frequencies of the yes/no distraction/anxiety items to estimate how many students experienced those symptoms. Next, we evaluated the bivariate correlation between students’ total digital competence scores and their total problematic smartphone use scores using Pearson’s *r*. This correlation provided a preliminary test of our hypothesis that digital competence is inversely related to problematic use. We then conducted a path analysis via structural equation modeling (SEM) to more rigorously test the directional relationship and to account for measurement error. In the SEM, digital competence was specified as an exogenous latent factor indicated by its multiple item parcels (we created 5 item parcels to represent the breadth of the DCQ-US domains), and problematic smartphone use was specified as an endogenous latent construct indicated by its subscale item parcels (we grouped the 15 PMPUQ-R items into 3 parcels reflecting the dependent, prohibited, and dangerous use dimensions). We allowed the two latent factors to covary and then added a structural path from digital competence to problematic use. The model was estimated using maximum likelihood. Model fit was evaluated with standard indices: chi-square (χ^2^), Comparative Fit Index (CFI), and Root Mean Square Error of Approximation (RMSEA), with criteria of CFI > 0.95 and RMSEA < 0.06 indicating good fit. We also tested for multigroup invariance across gender by comparing a model where the digital competence → problematic use path was free to differ between males and females versus one where it was constrained to be equal. A non-significant change in model fit (Δχ^2^) would suggest the effect does not differ by gender. The statistical significance level was set at *p* < 0.05 (two-tailed) for all analyses.

## 3. Results

### 3.1. Descriptive Statistics and Correlation

On average, the participating students exhibited moderate-to-high levels of problematic smartphone use; the mean PMPUQ-R score was 2.9 (SD = 0.6) on the 1–4 scale, indicating that many students reported engaging in problematic behaviors “sometimes” to “often”. Notably, 72% of the sample answered “yes” to experiencing frequent distraction or anxiety related to their phone (affirming at least one of the supplementary yes/no items about phone unavailability causing distress). This highlights the pervasiveness of phone-related preoccupations among these young adults. In terms of self-perceived digital skills, the students’ digital competence levels varied widely. The average DCQ-US score was 4.1 (SD = 0.5) on the 1–5 scale, suggesting that overall they felt quite confident in their digital abilities. Many students reported strength in basic technical tasks (e.g., installing apps, searching for information), whereas fewer felt very adept in more advanced areas like content creation or critical evaluation of online information.

Crucially, as hypothesized, digital competence was inversely related to problematic phone behavior, as shown in Table 1. The Pearson correlation between the total digital competence score and the total problematic smartphone use score was r = −0.38 (** *p* < 0.001), indicating a moderate negative association. In other words, students with higher levels of digital competence tended to have lower scores on smartphone addiction/problematic use measures. This finding is consistent with our expectation that digital competence can serve as a protective factor. While correlation does not imply causation, the strength of this relationship (approximately 14% of variance explained) is notable for psychosocial survey data. It suggests that those who are more skilled and mindful in the digital domain experience fewer symptoms of dependency or risky use with their phones. The negative association held in partial correlations controlling for age and gender as well (partial r = −0.35, *p* < 0.001). There was no evidence of a gender difference in this correlation: for both female and male students, greater digital competence was associated with lower problematic use to a similar degree.

### 3.2. Structural Equation Model

We next tested the hypothesized directional relationship using SEM. The structural model specifying Digital Competence → Problematic Mobile Use fit the data well. The model’s fit indices met recommended criteria: for example, the Comparative Fit Index (CFI) was 0.98, indicating an excellent fit, and the RMSEA was 0.031 (90% confidence interval [0.020, 0.041]), well below the 0.06 benchmark for close fit. The chi-square goodness-of-fit test was not significant (χ^2^(df = 112) = 125.4, *p* = 0.18), further suggesting the model adequately represents the covariance in the data.

Descriptive patterns showed that the most frequent smartphone habits among students included immediately checking notifications (84%), using social media during study periods (69%), extending phone use beyond planned time (71%), and multitasking with non-academic apps (63%). These behaviors were positively correlated with higher PMPUQ-R scores (r = 0.38, *p* < 0.001), providing contextual insight into students’ typical smartphone routines.

The main sample characteristics and descriptive statistics are presented in Table 2.

As illustrated in Figure 1, students’ digital competence had a significant negative effect on their level of problematic smartphone use in the structural model. The standardized regression weight was β = −0.42 (** *p* < 0.001), meaning that a one standard deviation increase in the latent Digital Competence factor was associated with a 0.42 standard deviation decrease in the latent Problematic Use factor. This path coefficient indicates a moderately strong protective influence. Digital competence accounted for about 18% of the variance in problematic use in the model. No other predictor variables were included in this core model, and for parsimony, we did not add age or gender as controls (given their non-significant bivariate relationships with the outcome). The SEM implicitly accounts for measurement error by using latent constructs, which likely contributed to the slightly stronger effect size (β = −0.42) compared to the raw correlation (−0.38).

To support the conceptual readability of the structural equation model presented below, we explicitly clarify the two central constructs depicted in Figure 2. Digital competence, as operationalized through the DCQ-US, refers to a multidimensional set of knowledge, technical skills, critical attitudes, and self-regulatory abilities that enable individuals to use digital technologies safely, responsibly, and effectively. It includes information literacy, communication and collaboration, digital content creation, safety, and problem-solving.

In contrast, problematic smartphone use refers to a psychological pattern of maladaptive engagement with a smartphone, characterized by dependence, loss of control, excessive use, and negative emotional or functional consequences. In this study, it was measured using the PMPUQ-R, which captures dependent use, prohibited use, and dangerous use.

The schematic model in Figure 2 visually represents the theoretically grounded negative association between these constructs: higher levels of digital competence are associated with lower levels of problematic smartphone use.

We conducted additional analyses to check the robustness of this finding; a multi-group SEM by gender revealed that the effect of digital competence was invariant between male and female students. Constraining the Digital Competence → Problematic use path to be equal for both genders did not worsen fit (Δχ^2^ was negligible, *p* > 0.50). In separate estimates, the path coefficient was β ≈ −0.40 for females and β ≈ −0.44 for males (both ** *p* < 0.001). Thus, there is no evidence that gender moderates the protective effect of digital competence in this sample—it appears beneficial for reducing problematic use in both groups. We also explored non-linear effects by adding a quadratic term for digital competence, but this effect was not statistically significant, suggesting the relationship is approximately linear within the observed range of competence.

### 3.3. Additional Analyses by Age and Gender

Supplementary analyses examined the association between digital competence and problematic smartphone use separately by gender (Figure 3).

In summary, our results support the hypothesis that digital competence is associated with lower problematic smartphone use among Paraguayan university students. The correlation and SEM analyses converge on the idea that students who are more adept and conscientious in their use of digital technologies experience fewer addictive behaviors or negative consequences related to smartphone use. This finding holds even when accounting for potential confounds and applies similarly across genders.

## 4. Discussion

This study examined whether digital competence can act as a psychological buffer against problematic smartphone use among Paraguayan university students, and the findings provide clear evidence in support of this role. As predicted, students with higher digital competence scores tended to report significantly fewer problematic behaviors with their mobile phones. In practical terms, the most digitally skilled and savvy students were less “addicted” to their smartphones—they experienced fewer feelings of panic or anxiety when separated from their device, engaged less in inappropriate or risky phone use (such as using phones in class or while driving), and generally exerted better control over their smartphone habits. The magnitude of the effect (β ≈ −0.4) suggests that digital competence is an important factor, though certainly not the only factor, in unhealthy phone use tendencies ([14]).

Although subscale-level analyses could not be performed with the anonymized dataset, the theoretical structure of the DCQ-US provides meaningful insight into how each dimension may differentially relate to problematic smartphone use. The critical and evaluative literacy dimension is expected to serve as the strongest protective factor, as students with higher evaluative skills are better able to identify manipulative digital design features, manage persuasive notifications, and regulate attention in digital environments. Safety and privacy management skills may also protect against maladaptive patterns by reducing anxiety and uncertainty related to notifications, personal data exposure, and digital risks. In contrast, technical-operational skills may have a weaker protective effect, as they reflect functional ability rather than self-regulatory competence. These theoretically grounded pathways align with recent research showing that reflective and self-regulatory aspects of digital competence—not merely operational skills—are the key factors associated with lower problematic smartphone use. This multidimensional interpretation reinforces our argument regarding the protective value of advanced, critical digital skills.

Our findings align with several theoretical perspectives and prior empirical studies; they echo results from a recent study on Turkish student-athletes, which found that higher digital competence was associated with lower phubbing behavior and better social well-being. In that context, athletes who were more competent with digital technology were less likely to compulsively check their phones and ignore real-life interactions, implying an ability to self-regulate and balance their phone use. Similarly, Tao et al. found that Hong Kong adolescents with greater digital competence were somewhat protected from the mental health harms of excessive screen time, partly by avoiding Internet addiction ([17]). These convergent findings from different populations support the idea that digital competence fosters resilience against the adverse outcomes of high-tech engagement. The concept of digital competence as a protective factor can be seen through the lens of empowerment: students who know how to use technology effectively and who understand its pitfalls may consciously limit deleterious behaviors (like doom-scrolling at 3 a.m. or texting while driving), much as health-literate individuals might make better dietary or exercise choices.

It is also worth noting that our results reinforce the value of looking at positive psychosocial characteristics (here, digital skills) in addiction research, rather than focusing exclusively on deficits or risk traits. Much of the smartphone addiction literature has centered on what types of people are more prone to overuse—for example, those with high impulsivity, loneliness, or stress tend to have higher problematic smartphone use ([10]). While that approach is valuable, our study suggests an alternative and complementary approach: enhancing certain competencies might reduce everyone’s risk, regardless of their predispositions. In our sample, the negative link between digital competence and problematic use held true even after controlling for gender and age, which hints at a broadly applicable benefit. This finding aligns with general models of digital well-being, which propose that improving users’ skills and awareness can lead to more intentional and balanced technology use.

Despite the clear trend observed, it is important to interpret our results in light of some cultural and contextual factors. Paraguay is a country where the digital transformation in education has accelerated only in recent years. Compared to students in more tech-saturated environments (e.g., East Asia or Western Europe), Paraguayan university students might have more to gain from increases in digital competence, as there may be a wider gap between those with higher versus lower competence. In a context where digital education is still developing, students who attain a high level of digital competence may also internalize more knowledge about the consequences of overuse and strategies for prevention. For example, being aware of apps that track screen time, or knowing how constant notifications can hijack attention. This awareness can translate into behaviors like turning off non-essential notifications, setting usage limits, or engaging in “digital detox” periods, which would naturally reduce problematic usage.

Another point of discussion is the nuanced nature of digital competence itself. Digital competence is a multifaceted construct, and not all components may guard equally against problematic use. Our measure encompassed technical skills (like installing software or troubleshooting devices) as well as higher-order skills (like evaluating information and understanding online privacy). It is conceivable that the critical and reflective dimensions of digital competence are doing the “heavy lifting” in reducing smartphone addiction, rather than the pure technical know-how. In other words, a student who is merely very tech-savvy (able to use many apps, etc.) but lacks critical self-regulation might actually engage more with their phone, potentially increasing risk—this scenario might describe the Egyptian finding where “digital media literacy” correlated positively with addiction scores. That study ([17]) measured digital competence in terms of media usage and technical knowledge, which could naturally correlate with more frequent phone use and thus more opportunities for addiction. By contrast, if digital competence is measured with an emphasis on reflective use and understanding consequences (as the DCQ-US does to some extent), higher scores should correlate with more prudent behavior, which is what we observed. Future research could delve deeper into which specific competencies (e.g., cybersecurity knowledge, content creation skills, information evaluation, or even digital well-being awareness) are most protective against problematic smartphone use. This would help tailor educational interventions to focus on the most impactful skill areas.

Our study also contributes region-specific insight by focusing on Latin American university students. The Latin American context presents both similarities and differences compared to other regions regarding smartphone use. Similar to reports from Asia and Europe, we found a high prevalence of phone-related anxiety (nomophobia) in Paraguayan students (nearly three-quarters felt anxious without their phones), underlining that youth from developing digital economies are just as attached to their devices as those elsewhere. On the other hand, the strong connection we observed between digital competence and reduced problematic smartphone use might reflect particular educational or cultural dynamics in Paraguay. It is possible that universities in our sample that emphasize technology training also inadvertently teach digital responsibility, or that students who seek digital skill development are more academically engaged and thus less likely to succumb to digital distractions. Recent transnational studies conducted across Latin America highlight significant variations in levels of digital competence and the necessity for developing specific plans for digital skills at the university level. Furthermore, preliminary evidence from neighboring countries is encouraging: a recent multicenter study across six Latin American nations recommended enhancing digital competence as part of strategies to address smartphone overuse and its health impacts on students ([7]). Our data provide empirical support for those recommendations, showing the direct association between digital competence and healthier usage patterns.

Our findings support the interpretation of digital competence as a psychological defense mechanism. Classical psychological theories describe defense processes as strategies that reduce exposure to emotional distress and impulsive behaviors ([19]; [5]). Students with higher digital competence showed fewer anxiety-related behaviors (e.g., discomfort when unable to check their smartphone) and fewer maladaptive habits such as compulsive checking or extended unplanned use. This pattern aligns with contemporary views of coping and self-regulation in digital contexts, suggesting that digital competence may buffer individuals from the emotional and behavioral dysregulation associated with problematic smartphone use ([4]; [6]).

RQ1 asked to what extent Paraguayan university students report problematic smartphone use.

Our findings show moderate to high levels of problematic smartphone use across the three PMPUQ-R dimensions (dependent, prohibited, and dangerous use), with a high prevalence of phone-related anxiety (72%), indicating that problematic smartphone behaviors are widespread in this population.

RQ2 examined the relationship between the different domains of digital competence and problematic smartphone use.

The results indicate a clear negative association between overall digital competence and problematic smartphone use. Students with higher digital competence scores tended to report fewer maladaptive phone behaviors, suggesting that digital competence is inversely related to problematic use patterns.

RQ3 investigated whether digital competence functions as a protective factor against problematic smartphone use.

The structural equation model confirmed that digital competence significantly and negatively predicts problematic smartphone use (β = −0.42, *p* < 0.001), supporting its role as a psychological protective factor among Paraguayan university students. Taken together, these findings provide a clear and coherent answer to the three research questions proposed in this study.

### 4.1. Limitations

We acknowledge several limitations of this research. First, the study’s cross-sectional design restricts our ability to infer causality. While our theoretical model posits that higher digital competence leads to lower problematic use, it is also plausible that students who reduce their phone addiction have more time and mental bandwidth to develop digital skills—or that an unmeasured third variable, such as general self-regulation or socio-economic status, influences both domains ([1]). Longitudinal and intervention studies using rigorous designs ([9]) would be valuable to clarify the temporal ordering and underlying mechanisms. Second, all measures were self-reported, which introduces the potential for common-method bias and social desirability effects. Past research shows students may overestimate their digital abilities or underestimate their problematic use due to self-perception ([15]). Including objective indicators in future studies—such as logged smartphone use data or standardized digital skill tests—would strengthen the validity of our findings.

Third, our sample, while sizeable, was not randomly selected and consisted predominantly of urban, tech-accessible university students. This raises the issue of generalizability, as our results may not extend to rural students, older adults, or broader Paraguayan populations. Self-selection bias may have occurred, with tech-savvy participants being more likely to respond ([1]). Although efforts were made to mitigate bias through broad recruitment and anonymity assurances, the limitation remains. Finally, cultural factors specific to Paraguay (e.g., collectivist values, academic norms, smartphone use culture) were not explicitly measured but could influence the observed relationships ([2]). Qualitative research could enrich our findings by exploring how students interpret “healthy” versus “unhealthy” phone use in their own context.

A further limitation concerns the absence of potentially relevant sociodemographic and academic covariates that could influence the association between digital competence and problematic smartphone use. Variables such as socioeconomic status, academic performance, access to digital resources, and prior digital training were not included in the present analyses. These factors may partially account for individual differences in both digital skill development and smartphone-related behaviors. Future studies would benefit from incorporating these covariates into multivariate models to better isolate the unique contribution of digital competence and to reduce the risk of residual confounding in the observed relationships.

### 4.2. Future Directions

Building on these findings, future research should employ intervention studies—such as implementing a digital competence workshop or curriculum and then tracking smartphone usage outcomes over time ([9]). If digital competence training demonstrably reduces problematic smartphone use, this would strongly justify universities and policymakers investing in such programs. Additionally, investigations should examine potential mediators (such as self-regulation, time management, or technology mindfulness) and explore how physical activity may interact with digital behaviors ([20]). Another worthwhile direction is to determine whether the protective effect of digital competence extends to other digital addictions—such as problematic social media use, online gaming, or more general internet addiction. Given the frequent co-occurrence of these behaviors, strengthening digital competence may offer broad benefits across multiple domains of digital life, a hypothesis that warrants verification ([2]).

To provide a deeper behavioral interpretation, we expanded the Results section by linking the statistical findings to concrete item-level behaviors assessed in the survey. The observed patterns suggest that anxiety-related behaviors (e.g., discomfort when unable to check the phone) coexist with habitual use patterns such as checking notifications reflexively or multitasking with social media during study periods.

Beyond the global negative association between digital competence and problematic smartphone use, our results align with specific behavioral manifestations captured in the survey. The descriptive analyses revealed clear patterns connecting students’ daily phone habits with their total PMPUQ-R scores. Behaviors such as immediately checking notifications upon receipt, extending phone use beyond intended time, or multitasking with social media during study periods were consistently associated with higher problematic use scores. These concrete behaviors provide an applied behavioral context to the quantitative findings, illustrating the mechanisms through which inadequate digital competence may contribute to maladaptive phone-related routines.

In addition, individual items of the DCQ-US shed light on why students with higher digital competence show fewer problematic behaviors. Higher-competence students reported greater ability to evaluate the credibility of online information, adjust privacy and notification settings, solve digital problems, and use digital tools in a planned and intentional manner. These skills likely enhance self-regulation, reduce impulsive device checking, and mitigate anxiety-driven behaviors (e.g., discomfort when unable to access the phone). As such, the survey items themselves help explain the psychological processes underlying the observed associations.

Taken together, the alignment between item-level behaviors (e.g., compulsive checking, extended usage, nomophobia-like discomfort) and broader PMPUQ-R dimensions (dependent use, prohibited use, and dangerous use) reinforces the behavioral validity of the constructs measured. This expanded interpretation strengthens the theoretical relevance and practical significance of the findings for behavioral science research.

Integrating digital competence training into university curricula can be concretely operationalized across disciplines through domain-specific applications. In education programs, students can be trained to critically evaluate online information sources, design safe and responsible digital learning environments, and model healthy smartphone habits for school-aged learners. In health sciences, digital competence can support the responsible use of mobile health applications, data privacy management, and the prevention of technology-related anxiety or dependence among patients. In engineering and computer science, students can engage in secure configuration of devices, ethical management of digital systems, and the design of user-focused technologies that minimize addictive patterns. In social sciences and psychology, curricular modules can focus on analyzing digital behaviors, interpreting smartphone-related data, and understanding the cognitive and emotional mechanisms underlying problematic use. These concrete curricular entry points demonstrate how digital competence can be meaningfully embedded across faculties, supporting the development of healthier and more intentional technology engagement among university students.

Future research should also examine whether the protective mechanisms of digital competence extend to other forms of digital dependence, such as problematic social media use, compulsive gaming, streaming overuse, or short-form video addiction. Theoretical frameworks such as the I-PACE model ([2]), which situates digital addiction within the interaction of personal predispositions, affective regulation, and executive control, offer a useful foundation for generating specific hypotheses. For example, it may be hypothesized that: (H1) higher digital competence predicts lower problematic social media use through improved self-regulation; (H2) evaluative and reflective digital skills mitigate compulsive gaming by enhancing awareness of persuasive design elements; and (H3) digital safety and privacy management skills reduce excessive engagement with short-form video platforms by limiting algorithm-driven exposure. Testing these hypotheses across different digital behaviors would clarify whether digital competence acts as a broad protective factor or whether its effects vary across specific digital environments.

## 5. Conclusions

This study demonstrates that digital competence acts as a protective factor against problematic smartphone use among Paraguayan university students. Paraguayan undergraduates who scored higher in digital competence skills and awareness were substantially less prone to exhibiting signs of smartphone addiction and related dysfunctional behaviors. This finding is heartening, as it suggests that bolstering students’ digital competence and an achievable educational goal may help inoculate them against the pitfalls of excessive smartphone use.

These results reinforce the notion that digital competence may operate as a psychological defense that mitigates dependence-like behavior patterns and supports healthier engagement with smartphones. Strengthening digital competence could therefore serve not only as an educational goal but also as a preventive psychological strategy aligned with established theories of coping and behavioral regulation.

Our results carry an optimistic practical message: efforts to promote digital competence could double as interventions to improve young people’s digital well-being. Universities and educational stakeholders should consider incorporating digital competence training into curricula, not only to enhance academic and career skills, but also to encourage healthier relationships with technology. By learning how to critically evaluate online content, manage privacy settings, balance online-offline life, and recognize the tactics that apps use to capture attention, students can become more empowered, self-regulated users rather than passive subjects of their devices. In an era when smartphones are deeply embedded in daily life, nurturing a generation of digitally competent users may be one of the most constructive strategies to curb technology’s adverse effects.

Fostering digital competence is one of the most practical and promising keys to promoting healthier technology engagement among students.

## Figures and Tables

**Figure 1 behavsci-15-01687-f001:**
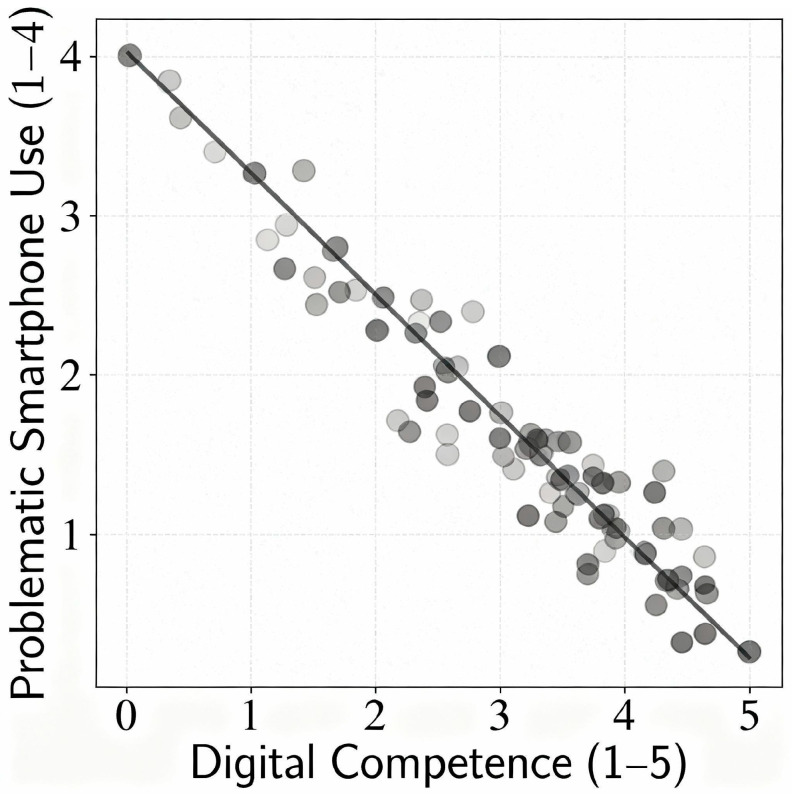
Scatterplot illustrating the negative relationship between digital competence and problematic smartphone use in the sample. Each point represents a student, plotted by their Digital Competence score (x-axis, 1–5 scale) and Problematic Use score (y-axis, 1–4 scale). A linear trend line (dashed) is shown, indicating that higher digital competence is associated with lower problematic use (Pearson r = −0.38, *p* < 0.001).

**Figure 2 behavsci-15-01687-f002:**
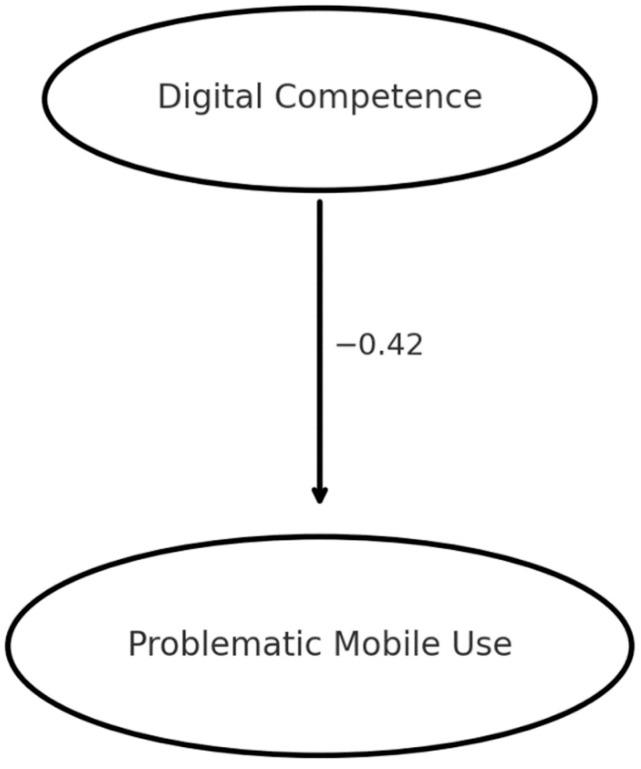
Conceptual structural model illustrating the negative association between digital competence and problematic smartphone use. The diagram provides a simplified visual representation of the standardized path (β = −0.42), helping readers interpret the central relationship tested in the SEM model.

**Figure 3 behavsci-15-01687-f003:**
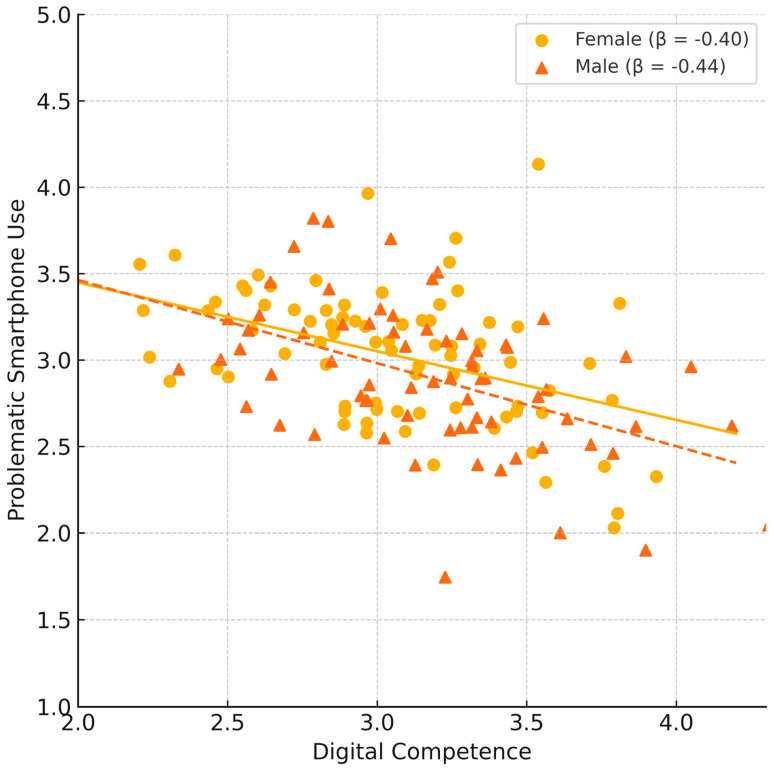
Scatterplot illustrating the relationship between digital competence and problematic smartphone use by gender. Higher digital competence is associated with lower problematic smartphone use for both females (β = −0.40) and males (β = −0.44).

**Table 1 behavsci-15-01687-t001:** Summary of Sample Characteristics and Key Measures in the Structural Model.

Aspect Analyzed	Description/Approach
Model Estimation Method	Maximum likelihood estimation for the structural equation model
Model Fit Indices Used	Chi-square (χ^2^), Comparative Fit Index (CFI), Root Mean Square Error of Approximation (RMSEA)
Accepted Criteria for Good Fit	CFI ≥ 0.95, RMSEA ≤ 0.06
Multigroup Invariance Analysis	Tested gender differences by comparing unconstrained and constrained models
Statistical Significance Level	Statistical significance at *p* < 0.05 (two-tailed)
Measurement Parcels	Items grouped by subscales for both digital competence and problematic smartphone use constructs
Path Analysis	Structural path added from digital competence to problematic use

The table summarizes the main descriptive values of the sample and the key variables analyzed in the structural model.

**Table 2 behavsci-15-01687-t002:** Summarizes the sample characteristics and key measures (N = 500).

Statistic/Measure	Value
Mean age (SD)	21.4 years (SD = 2.7)
Gender distribution	61% Female, 39% Male
Frequent phone distraction/anxiety ^1^	72% of students (answered “yes”)
Digital Competence (DCQ-US) mean score	4.1 (SD = 0.5) on 1–5 Likert scale
Problematic Phone Use (PMPUQ-R) mean score	2.9 (SD = 0.6) on 1–4 Likert scale
Correlation: Digital Competence vs. Problematic Use	*r* = −0.38 (** *p* < 0.001)

^1^ Percentage of participants who answered “yes” to experiencing frequent distraction or anxiety due to not having access to their smartphone.

## Data Availability

The data presented in this study are available on reasonable request from the corresponding author. The data are not publicly available due to privacy and ethical restrictions.

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
