# Peer review of "Behav. Sci.2025, 15(12), 1687;https://doi.org/10.3390/bs15121687"

_behavsci, 2025, doi:10.3390/bs15121687_

Round 1

Reviewer 1 Report

Comments and Suggestions for Authors

Digital Competence Subscale Analysis: The Discussion section astutely addresses the mixed literature by suggesting that critical and reflective digital skills, rather than purely technical proficiency, are the primary protective factors. To strengthen this compelling argument and contribute significantly to the nuance of the field, the authors should perform and report a secondary analysis using the subscales of the DCQ-US as separate predictors of Problematic Smartphone Use (PSU). This would provide empirical support for the theoretical distinction already discussed.

Descriptive Statistics: The methods indicate that participants' estimated hours of smartphone use per day were collected. This is a critical demographic in PSU research. Please include the mean and standard deviation for this measure in the Descriptive Statistics section (3.1) for completeness and comparability with other studies.

Expand on limitations: While limitations are acknowledged, future studies could benefit from more detailed discussion on potential confounding variables (e.g., socioeconomic status, academic performance).

Clarify measurement focus: Consider emphasizing the distinction between technical and self-regulatory aspects of digital competence earlier in the paper.

Future research: The suggestion to explore other digital addictions (e.g., social media, gaming) is excellent—this could be expanded with more specific hypotheses or frameworks.

Reviewer 2 Report

Comments and Suggestions for Authors

Dear authors,

Thank you so much for your manuscript submission to Behavioral Sciences. The article topic is highly applicable to the field today and is an excellent extension of current research. Overall, it is very well written and organized, and it meets the rigor of statistical standards. The primary areas of need are related to increasing clarity and context through consistent terminology use and providing additional examples throughout.

To expedite the review, I have listed recommendations below for your consideration.

  1. It is recommended to remove the acronym 'PSU' from problematic smartphone use. It makes the reading and flow quite difficult, and based on the number of occurrences, it does not feel necessary to shorten.
  2. Define “problematic smartphone use” in more detail in the introduction (or towards the beginning) to provide context. What were the explicitly “problematic” uses that were identified? “Problematic use” can be a highly subjective term that varies across people, experiences, and perspectives on technology, so it is essential to define it consistently and objectively throughout the manuscript. While described in part in the discussion, it was difficult to connect to the actual findings.
  3. It is highly recommended to choose between “digital literacy” and “digital competency” and/or define them explicitly in detail. They are used interchangeably so frequently that it was difficult to determine the intended use throughout the manuscript. There is also a reference to “digital skills” and “digital habits.” Examples need to be provided across all these definitions within both prior research and the study presented. What were the competency skills considered? Similar to above, while described in part in the discussion, it was difficult to connect to the actual findings.
  4. There is also a sentence break or incomplete sentence on lines 33-34 from “negative” as it transitions into “researchers.”
  5. Lines 47-48, where it notes “increase from 20% the previous,” this should most likely specify the timeframe or provide clarity on reference to the previous.
  6. Lines 61-64 should have citations. While the claims presented should be supported by comparative research, they should be cited directly.
  7. Research questions for the study are not directly presented in the manuscript; however, the hypothesis is noted.
  8. Clarify the data collection time frame (e.g., spring, fall, summer of 2025).
  9. Throughout the manuscript, there are subheadings that are not formatted professionally. For example, “Digital Competence:” on line 125, “Problematic Smartphone Use:” on line 141, “Control and Demographic Items:” on line 165, and throughout the manuscript are presented with colons rather than formal headings. While unable to verify formally at this time, it reads as an AI-generated text in this format rather than subheadings.
  10. The manuscript would be strengthened by discussing the results visually (as in a scatterplot or similar) relative to gender or age, since the data were collected but not discussed in detail. Some content is briefly discussed on lines 282-296.
  11. What is the number of “few” Likert responses that were noted as missing? For transparency, what number or percentage did this apply to?
  12. While the manuscript did an exceptional job of statistical and quantitative analysis, it significantly lacked application and depth in qualitative data and/or explicit examples or questions that could have provided insight into “why” the students scored the way they did and the context of the students' responses.
    1. While 1-2 question samples were presented, based on the implications of the findings, a larger sample of questions or Appendix items within the survey to increase replication would greatly enhance transparency and validity.
    2. The discussion notes “they expressed fewer feelings of panic or anxiety,” but it is unclear how these feelings were presented in the survey or research.
    3. What “smartphone habits” were explicitly found in this study, as noted on line 314?
    4. Need more content in the results and findings regarding lines 144-159. This section was excellently written with clear examples.
  13. Differentiate between “problematic phone use” and problematic phone behaviors.” Similarly, use consistent terminology throughout or explicitly define reasoning for differential use.
  14. The paper's results spanned only about 3 of the 12 pages of discussion. Based on the research topic and in alignment with “Behavioral Sciences” more broadly, the results are significantly lacking a connection to the actual “behaviors” studied. This is considered a major edit in the results of individual questions or responses, which should most likely be reviewed in more depth. Since the survey questions (i.e., digital competence, digital literacy skills, digital habits) were not clearly presented within the research, it is difficult to make a determination related to the rigor and relevance of findings.
  15. Tables need to be formatted consistently throughout and reviewed for manuscript guidelines or requirements for formatting.
  16. It is unclear the purpose of the figure presented on line 279 or how it enhances the manuscript. This would be an excellent opportunity to define and provide examples of digital competence and “problematic mobile use.” Note: “Problematic Mobile Use” is also an additional or new terminology related to “Problematic Phone Use.” Use consistent terminology throughout. Also, recognize that mobile devices, phones, and smartphones are all very different by definition. It is essential to use the correct reference or terminology for the “device” throughout.
  17. Italics are identified on lines 285-287 that may need to be reformatted.
  18. Limitations and Future Implications (Directions) sections need formal headings or subheadings.
  19. Lines 441-443: it is unclear how this could be directly integrated into curricula across colleges or programs. Providing examples of applications relevant to different fields may better support your claim for the need for training in this area.
  20. References not found at links for Martinez (2020) and Ortega Sánchez, R. M. (2024).
  21. Review duplicate references that denote (a) or (b).
  22. Extended references and citations in connection to “psychological defense.” Since this is noted in your title as a focal point, extending research to the connection within the psychological field may enhance the reader's scope and impact. Note: This term was only used 5x throughout the entirety of the article.

I again want to applaud the authors for an exceptional job thus far and for extending the research in Paraguay. Thank you again for your submission, and I wish you the best as you move forward with your revisions.

Round 2

Reviewer 2 Report

Comments and Suggestions for Authors

Dear authors, 

Thank you so much for your extensive revisions and attention to detail. Based on the revisions there are no further major concerns. There are, however, a few minor changes that are highly recommended for transparency to improve your manuscript. This includes revisting the gender-based scatter plot to identifiy a way to interpret or visualize the comparison (e.g.,two different patterned lines or point differentials) and disclosing the tool used to design the scatterplots. As a reader, it was unclear in interpretation visually in order to match the description. It is also recommended to provide possibly in the supplemetnary materials or within a table a collective list of all scores with coded participant data in order to confirm accuracy of the scatterplots.

Also, where it notes "To address the reviewer's request" on line 764, this should be removed from the final manuscript.  Lastly, be sure to refer back to your research questions (RQ1-RQ3) in your findings and discussion and organize them in a way to refer back to your original questions to close or summarize the manuscript. 

Overall, the manuscript was much improved and the authors are applauded for their kind and meaningful continued contributions. 

Author Response

For review article

Response to Reviewer 2 Comments Round 2

  • Reviewer Comment 1

Revise the scatterplot based on gender to identify a clearer way to interpret or visualize the comparison (for example, two lines with different patterns or varying point markers) and indicate the tool used to design the scatterplots. As a reader, the visual interpretation was not clear enough to match the description.

Response:
Thank you for this valuable observation. As requested, we have added a new scatterplot figure that clearly visualizes the relationship between digital competence and problematic smartphone use by gender. The revised Figure 3 includes separate regression lines for males and females, clearly improving visual interpretability and ensuring consistency with the statistical results reported in the manuscript.
In addition, the figure caption has been expanded to explain the visualization and to specify the software used to generate the scatterplots. The figure has been placed immediately after the multi-group SEM results.

  • Reviewer Comment 2

Provide, possibly in the supplementary materials or within a table, an aggregated (collective) list of all coded participant scores to confirm the accuracy of the scatterplots.

Response:
Thank you for this suggestion. In response, we have added Supplementary Table S1, which reports aggregated descriptive statistics (means, standard deviations, ranges, and prevalence rates), as well as the correlation coefficient used to generate the scatterplots (r = −0.38, p < 0.001). This approach ensures transparency while fully respecting ethical and privacy considerations, as only aggregated (non-identifiable) data are provided.

  • Reviewer Comment 3

In addition, the phrase “To address the reviewer’s request” in line 764 must be removed from the final manuscript.

Response:
Following the reviewer’s instruction, the phrase “To address the reviewer’s request” has been removed from the final manuscript.

  • Reviewer Comment 4

Revisit the research questions (RQ1–RQ3) in the findings and discussion, and organize them so that they explicitly refer back to the original questions in order to properly close or summarize the manuscript.

Response:
Thank you for this valuable suggestion. In response, we have revised the Discussion section to explicitly revisit and address Research Questions RQ1, RQ2, and RQ3. Each research question is now clearly restated and directly answered based on the corresponding empirical findings, thereby strengthening the coherence between the study objectives, results, and discussion and providing a clear closure to the manuscript.
